# MULTI-MODAL POINT CLOUD COMPLETION WITH INTRA- AND INTER-GRAPH TRANSFORMER

## ABSTRACT

Multi-modal point cloud completion aims to leverage complementary image information to assist point cloud completion. Existing multi-modal approaches predominantly employ Transformers to facilitate interactions between different modalities. However, fully-connected attention-based Transformers lead to high computational cost and redundancy, and often fail to fully capture the complex relations between these modalities. To address these issues, we propose the **Intra**- and **Inter-Graph** Trans**former** ($\mathbf{I^2GraphFormer}$), which leverages sparse graph connections to restrict attention to neighboring nodes both within and across modalities. $I^2GraphFormer$ enhances interactions in terms of efficiency and expressiveness. Specifically, we model relations from both intra-graph and inter-graph perspectives, obtaining more expressive representations and producing higher-quality completion results. Extensive quantitative and qualitative experiments demonstrate that $I^2GraphFormer$ outperforms state-of-the-art multi-modal approaches across various evaluation scenarios with low complexity.

## 1 INTRODUCTION

Benefiting from the convenience of acquisition and storage, point clouds have progressively emerged as the principal data modality in 3D computer vision, thereby garnering increasing attention from researchers in the field. However, point cloud data collected from real-world environments are generally sparse and incomplete due to the limitations of 3D sensors and environmental conditions Fei et al. (2022); Tesema et al. (2024), which can adversely affect the performance of downstream tasks such as point cloud classification Liang et al. (2023), segmentation Du et al. (2024b); Guo et al. (2021), and registration Mu et al. (2024); Zhang et al. (2023b).

Single-modal point cloud completion Yuan et al. (2018) has emerged as an effective solution to this challenge, aiming primarily at reconstructing incomplete observations into complete and accurate 3D objects. Most methods Zhou et al. (2022); Wang et al. (2024) tend to design end-to-end networks to model the mapping between incomplete observations and complete shapes.

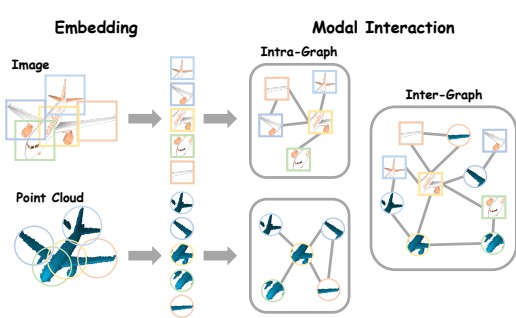

Figure 1: An illustration of our main idea. Firstly, the point cloud and image are encoded into local tokens during the embedding stage. Subsequently, these tokens are treated as nodes, and modal interaction is conducted from two perspectives, i.e., intra-graph (within each modality, namely point cloud and image) and inter-graph (between the point cloud and image modalities).

In addition, some approaches Huang et al. (2020); Alliegro et al. (2021); Yu et al. (2021) focus on employing autoencoders to recover the missing shapes from incomplete observations, rather than generating fully completed outputs. Although these single-modal methods have demonstrated certain performance, relying solely on point cloud data to learn the mapping from incomplete observa-

tions to complete objects is challenging due to the inherent incompleteness of the input point clouds Zhang et al. (2021); Xu et al. (2024).

Multi-modal point cloud completion Zhang et al. (2021), which introduces additional image information to assist point cloud completion, is proposed to address this challenge. This approach is particularly relevant as the development of cross-modal sensors promotes the use of multi-modal data to perceive the surrounding environment in fields such as autonomous driving Geiger et al. (2012) and transportation infrastructure Yan et al. (2024). Most existing multi-modal methods Aiello et al. (2022); Du et al. (2024a); Xu et al. (2024) employ Transformer-based Vaswani et al. (2017) architectures for modality fusion. However, traditional Transformers employ fully-connected attention mechanisms Sun et al. (2023), which lead to high computational cost and redundancy, and frequently fail to adequately capture the complex relations between these modalities.

To address these challenges, as shown in Figure 1, we propose $I^2$GraphFormer, which enhances modality interactions through sparse graph-based attention mechanism. First, the image and point cloud inputs are encoded into corresponding 2D and 3D tokens. We then construct separate graph structures within each modality, where Transformers model intra-graph relations and facilitate information propagation among nodes. Next, the point cloud and image tokens are jointly represented as nodes in a bipartite graph to capture inter-graph relations across modalities. Inter-graph Transformers are employed again to enable complementary information exchange between the two modalities. Finally, the enriched point cloud and image tokens are progressively upsampled through multiple dual-view guided upsampling modules to reconstruct a complete point cloud, resulting in more expressive representations and higher-quality completion results. Benefiting from the utilization of image information during the decoding stage, the final completed results contain richer details. Comprehensive quantitative and qualitative evaluations show that $I^2$GraphFormer surpasses state-of-the-art multi-modal methods across diverse assessment scenarios while sustaining low computational complexity. The main contributions of this paper are presented as follows.

- We propose $I^2$GraphFormer for multi-modal point cloud completion, demonstrating its high-efficiency completion capability in both synthetic and real-world scenarios.

- We propose $I^2$GraphFormer which deeply captures the local structures and feature relations within point cloud and image modalities from an intra-graph perspective. Furthermore, it effectively models the complex geometric and semantic correlations between point clouds and images, enabling complementary information exchange and cross-modal fusion.

- We propose a dual-view guided upsampling module (DVGUM) that directs the reconstruction process from both geometric and image perspectives, enabling the generation of finer-grained point clouds.

## 2 RELATED WORK

### 2.1 SINGLE-MODAL POINT CLOUD COMPLETION

Yuan et al. Yuan et al. (2018) have proposed a point completion network (PCN) that has pioneered research in single-modal point cloud completion. Building upon PCN, TopNet Tchapmi et al. (2019) utilizes a tree-structured decoder architecture to effectively generate accurate completion results. PF-Net Huang et al. (2020) employs a multi-scale encoder-decoder architecture to produce more detailed and refined completion results. MSN Liu et al. (2020) utilizes multiple decoders designed to generate diverse shape variations of the completed outputs. GRNet Xie et al. (2020) employs 3D grid representations to effectively model point clouds for single-modal point cloud completion. PoinTr Yu et al. (2021) is the first to apply the Transformer architecture to single-modal point cloud completion. SDT Zhang et al. (2023a) designs a skeleton-detail Transformer to generate fine-grained details in the completion results. SeedFormer Zhou et al. (2022) designs a seed-based upsampling Transformer to achieve detail preservation and shape reconstruction. PointAttN Wang et al. (2024) designs a Transformer composed entirely of attention mechanisms for point cloud completion. Moreover, recent methods such as SDS-Complete Kasten et al. (2023), ComPC Huang et al. (2025), and GenPC Li et al. (2025) have concentrated on applying the zero-shot paradigm to point cloud completion.

## 2.2 Multi-modal Point Cloud Completion

Recently, research on multi-modal point cloud completion, which involves using a partial point cloud and a complete image as input, has been thriving. As a pioneer, Zhang et al. Zhang et al. (2021) introduce a view-guided point cloud completion (ViPC) framework, laying the groundwork for subsequent advancements in multi-modal point cloud completion. Based on the ViPC framework, CSDN Zhu et al. (2024) integrates point clouds and images by designing an adaptive normalization operator, effectively merging global shapes from the image modality to the point cloud modality. Inspired by the success of Transformer Vaswani et al. (2017) in natural language processing (NLP) and computer vision (CV), several approaches Aiello et al. (2022); Du et al. (2024a); Xu et al. (2024) have been developed that utilize Transformer-based networks for multi-modal point cloud completion.

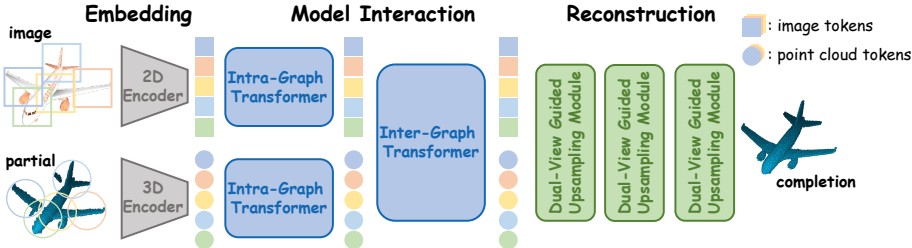

Figure 2: Overview of I²GraphFormer consists of embedding, modal interaction, and reconstruction three parts. Point cloud and image are first encoded into tokens through 3D and 2D encoders, respectively. These tokens are then further processed separately by two independent intra-graph Transformers to obtain refined tokens. Subsequently, the refined tokens are jointly fed into an inter-graph Transformer to enable mutual complementation. Finally, the complemented tokens are decoded into a completion result via a stacked dual-view guided upsampling module.

## 3 Method

### 3.1 Overview

Figure 2 illustrates an overview of I²GraphFormer. The input, consisting of a point cloud and an image, is respectively encoded into local tokens through PointNet++ Qi et al. (2017) and ResNet He et al. (2016). Then, the local tokens of the point cloud and image are further explored through separate intra-graph Transformers to capture their internal information. Subsequently, the local tokens of the point cloud and image are processed by an inter-graph Transformer to explore cross-modal information and enable mutual complementarity between the two modalities. Finally, the complemented tokens of the point cloud and image are jointly passed through three stacked dual-view guided upsampling modules to generate the final completion result.

### 3.2 Intra- and Inter-Graph Transformer

Existing methods mostly employ cross-attention mechanisms to achieve fusion of the two modalities. However, this approach typically models cross-modal interactions as a fully connected graph, where all nodes between the two modalities are intercon-

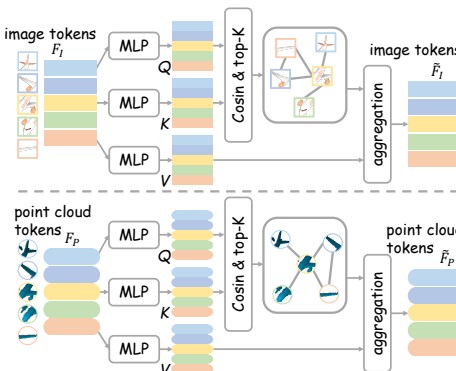

Figure 3: Overview of the Intra-Graph Transformer. An illustration of the image modality is shown above, with the point cloud modality depicted below.

nected. Intuitively, whether within the same modality (that is, within the point cloud or within the image) or across modalities (that is, between the point cloud and the image), **not all local tokens are similar and therefore do not all have edges connecting them**. Next, we will focus on our core contribution with novel graph Transformers for multi-modal point cloud completion.

**Intra-Graph Transformer.** Figure 3 illustrates the workflow of the intra-graph Transformer applied separately to the point cloud and image modalities. To present this part more clearly, we first denote the image tokens and point cloud tokens obtained from the 2D and 3D encoders as $F_I = \{f_i^I \mid i = 1, 2, \ldots, M\}$, where each $f_i^I \in \mathbb{R}^C$ and $F_P = \{f_j^P \mid j = 1, 2, \ldots, N\}$, where each $f_j^P \in \mathbb{R}^C$, respectively. The inputs of the intra-graph Transformers for the image and the point cloud are $F_I$ and $F_P$, respectively. Taking the intra-graph Transformer of the image modality as an example (**top half of Figure 3**), the image tokens are first passed through three MLPs to obtain query (Q), key (K), and value (V).

$$
\begin{cases}
Q = \mathrm{MLP}(F_I), \\
K = \mathrm{MLP}(F_I), \\
V = \mathrm{MLP}(F_I),
\end{cases}
\tag{1}
$$

where $Q = \{q_i\}_{i=1}^M$, $K = \{k_i\}_{i=1}^M$, and $V = \{v_i\}_{i=1}^M$. Then, the cosine similarity is computed between every pair of vectors in query and key to obtain similarity matrix. Next, based on the similarity matrix, for each query vector, we select its top K most relevant key vectors to obtain a sparser similarity matrix, which serves as the adjacency matrix. The edges between the $i$-th query vector and its top K most relevant value vectors are defined as

$$
a_{ij} = \frac{q_i \cdot k_{\mathrm{idx}_j}{}^T}{\|q_i\|_2 \cdot \|k_{\mathrm{idx}_j}\|_2}, j = 1, 2, \ldots, \mathrm{K}.
\tag{2}
$$

It is worth noting that each index $\mathrm{idx}_j$ belongs to the set $\{1, 2, \cdots, M\}$ for all $j = 1, 2, \ldots, \mathrm{K}$. Finally, we use this sparse adjacency matrix to perform information aggregation among the image tokens.

$$
\tilde{f}_i^I = \sum_{j=1}^{\mathrm{K}} (a_{ij} \odot v_{\mathrm{idx}_j}), i = 1, 2, \ldots, M,
\tag{3}
$$

where $\tilde{F}_I = \{\tilde{f}_i^I \mid i = 1, 2, \ldots, M\}$ denotes the output of intra-graph Transformer and each $\tilde{f}_i^I \in \mathbb{R}^C$. The intra-graph Transformer for the point cloud modality (**bottom half of Figure 3**) operates in a manner similar to the one described above. In short, the input is $F_P = \{f_i^P \mid i = 1, 2, \ldots, N\}$ and each $f_i^P \in \mathbb{R}^C$, and the output is $\tilde{F}_P = \{\tilde{f}_i^P \mid i = 1, 2, \ldots, N\}$ and each $\tilde{f}_i^P \in \mathbb{R}^C$.

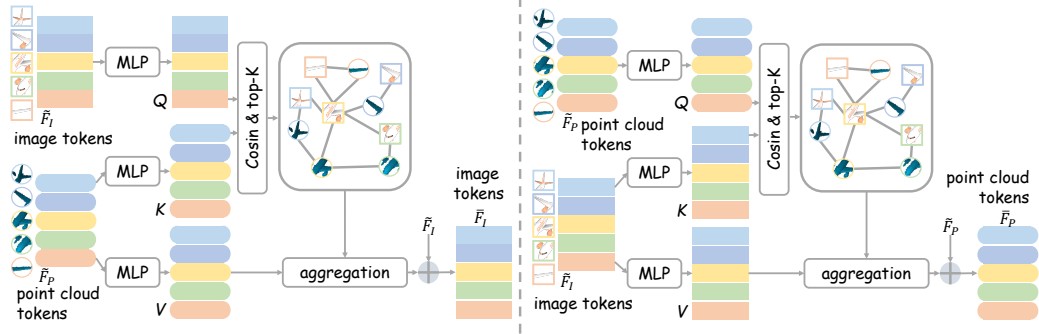

Figure 4: Overview of the Inter-Graph Transformer. The illustration on the left shows the point cloud modality supplementing the image modality, while the illustration on the right depicts the image modality supplementing the point cloud modality.

**Inter-Graph Transformer.** Figure 4 illustrates the process of the inter-graph Transformer facilitating the exchange of complementary information between the point cloud and image modalities. The left part of the figure depicts the point cloud supplementing the image modality, while the right part shows the image modality supplementing the point cloud.

For the sake of concise notation, we denote the image tokens and point cloud tokens input to the inter-graph transformer as $\tilde{F}_I$ and $\tilde{F}_P$, respectively. Taking the inter-graph Transformer that employs image tokens to supplement point cloud modality as an example (**right half of Figure 4**), the point cloud tokens and image tokens are first passed through three MLPs to obtain query (Q), key (K), and value (V).

$$\begin{cases} Q = \text{MLP}(\tilde{F}_P), \\ K = \text{MLP}(\tilde{F}_I), \\ V = \text{MLP}(\tilde{F}_I), \end{cases} \tag{4}$$

where $Q = \{q_i\}_{i=1}^N$, $K = \{k_i\}_{i=1}^M$, and $V = \{v_i\}_{i=1}^M$. Then, the cosine similarity is calculated between every query and key vector pair to form a similarity matrix. Based on this matrix, for each query vector, the top K most relevant key vectors are selected to create a sparser similarity matrix, which functions as the adjacency matrix. The edges between the $i$-th query vector and its top K most relevant value vectors are defined as

$$a_{ij} = \frac{q_i \cdot k_{\text{idx}_j}^T}{\|q_i\|_2 \cdot \|k_{\text{idx}_j}\|_2}, j = 1, 2, \ldots, \text{K}. \tag{5}$$

Notably, each index $\text{idx}_j$ belongs to the set $\{1, 2, \cdots, M\}$ for all $j = 1, 2, \ldots, \text{K}$. Finally, we use this sparse adjacency matrix to supplement point cloud tokens with image informations.

$$\bar{f}_i^P = \tilde{f}_i^P + \sum_{j=1}^{\text{K}} (a_{ij} \cdot v_{\text{idx}_j}), i = 1, 2, \ldots, N, \tag{6}$$

where $\bar{F}_P = \{\bar{f}_i^P \mid i = 1, 2, \ldots, N\}$ denotes the output of inter-graph Transformer (**right half of Figure 4**) and each $\bar{f}_i^P \in \mathbb{R}^C$.

The principle of using the inter-graph Transformer to supplement image tokens with point cloud information (**left half of Figure 4**) is similar to the description above.

Overall, the intra-graph Transformer first thoroughly explores the geometric information within the point cloud and the semantic information within the image, after which the inter-graph Transformer facilitates bidirectional information supplementation between the two modalities.

## 3.3 DUAL-VIEW GUIDED UPSAMPLING MODULE

The supplemented point cloud tokens $\bar{F}_P$ are first restored into a coarse point cloud $P_0$ through a stacked MLPs, which is omitted from the Figure 2 for simplicity. Then, as shown in Figure 5, we present the workflow of the DVGUM. Assuming the input consists of a low-resolution point cloud $P_{\text{in}}$, point cloud tokens $\bar{F}_P^{\text{in}}$, and image tokens $\bar{F}_I$, the output is a high-resolution point cloud $P_{\text{out}}$ along with updated point cloud tokens $\bar{F}_P^{\text{out}}$. The geometry guided branch constructs a top-K graph structure based on the point cloud coordinates $P_{\text{in}}$, and subsequently aggregates the point cloud tokens $\bar{F}_P^{\text{in}}$ using this graph structure to obtain refined point cloud tokens $\bar{F}_P^{\text{out}}$. The image-guided part is an inter-graph Transformer that aggregates point cloud information based on image features, **similar to the left half of Figure 4**. The point cloud features $\bar{F}_P^{\text{out}}$ obtained from geometric information (spatial coordinates) are

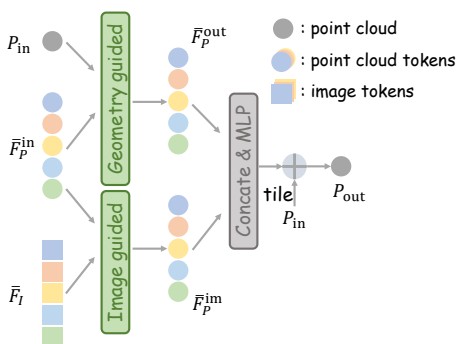

Figure 5: Overview of the Dual-View Guided Upsampling Module.

concatenated with the point cloud features $\bar{F}_P^{\text{im}}$ aggregated under the guidance of image information, and then passed through an MLP to learn the offsets for the high-resolution point cloud $P_{\text{out}}$.

$$P_{\text{out}} = \text{tile}(P_{\text{in}}) + \text{MLP}(\bar{F}_P^{\text{out}} || \bar{F}_P^{\text{im}}), \tag{7}$$

where $\text{tile}(\cdot)$ denotes copy operation and $||$ denotes concatenate operation. As shown in Figure 2, our method comprises three DVGUMs, which generate three progressive completion results at different stages, denoted as $P_1$, $P_2$, and $P_3$, respectively.

In summary, DVGUM generates local details and global shapes of high-resolution point clouds through a geometric perspective (point cloud spatial coordinate information) and an image perspective (supplemented image tokens as guidance).

### 3.4 LOSS

We adopt the Chamfer distance (CD) as the loss function to train the I$^2$GraphFormer. The formula for the CD is as

$$\mathcal{L}_{\text{CD}}(P, \hat{P}) = \frac{1}{|P|} \sum_{p \in P} \min_{\hat{p} \in \hat{P}} \|p - \hat{p}\|_2^2$$
$$+ \frac{1}{|\hat{P}|} \sum_{\hat{p} \in \hat{P}} \min_{p \in P} \|\hat{p} - p\|_2^2. \tag{8}$$

where $P$ and $\hat{P}$ are the completion result and ground truth, respectively. CD is commonly used to quantify the similarity between two point clouds. Since I$^2$GraphFormer utilizes a multi-stage decoding approach, it produces completion results at multiple resolutions, denoted as $P_0$, $P_1$, $P_2$, and $P_3$. The final loss is computed between the ground truth $P_{gt}$ and the completion result at each resolution, with the specific formulation as

$$\mathcal{L} = \sum_{i=0}^{3} \mathcal{L}_{\text{CD}}(P_i, P_{gt}). \tag{9}$$

## 4 EXPERIMENTS AND ANALYSES

### 4.1 DATASETS

**ShapeNet-ViPC.** The ShapeNet-ViPC dataset Zhang et al. (2021) is widely regarded as a benchmark dataset in the field of multi-modal point cloud completion. It comprises 13 categories, including airplane, cabinet, car, chair, lamp, sofa, table, watercraft, bench, monitor, speaker, fireman, and cellphone, with a total of 38,328 samples. Each sample is captured from 24 distinct viewpoints and consists of a complete image, an incomplete point cloud, and a corresponding complete point cloud. The experimental protocol adopted in this study is consistent with that of prior multi-modal approaches Xu et al. (2024) to ensure comparability and methodological coherence.

**KITTI.** The KITTI dataset Geiger et al. (2012) is acquired from real-world environments. The data utilized in this study are derived from the work of Cross-PCC Wu et al. (2025). Specifically, the dataset comprises 156 samples, each containing an incomplete point cloud alongside a corresponding complete image. The size of the image is 224×224.

### 4.2 EXPERIMENTAL SETTINGS

**Metrics.** Consistent with previous multi-modal methods Zhang et al. (2021); Zhu et al. (2024); Aiello et al. (2022); Du et al. (2024a); Xu et al. (2024), we use CD (multiplied by 1000) and F-Score as evaluation metrics. The F-Score is the harmonic mean of precision and recall. Precision evaluates the proportion of points in the predicted point cloud that are near the ground-truth point cloud, whereas recall measures the proportion of points in the ground-truth point cloud that are captured by the predicted point cloud.

**Implementation Details.** All experiments are performed on an NVIDIA RTX A6000. We use the Adam optimizer with an initial learning rate of 0.001, decayed by 0.7 every 40 epochs, and a batch size of 64. Following previous methods Zhu et al. (2024); Xu et al. (2024) on the ShapeNet-ViPC dataset, we perform supervised training and testing on 8 categories, and conduct generalization experiments on the remaining 4 categories. Since the KITTI dataset lacks ground truth labels, we train on the car category of the 3D-EPN dataset created by Wu et al. Wu et al. (2025), and test on the corresponding KITTI dataset they constructed.

| Methods | CD ↓ / F-Score ↑ | | | | | | | | |
|---|---|---|---|---|---|---|---|---|---|
| | Avg | Airplane | Cabinet | Car | Chair | Lamp | Sofa | Table | Watercraft |
| Single-modal Methods | | | | | | | | | |
| AtlasNet (CVPR 2018) | 6.062 / 0.410 | 5.032 / 0.509 | 6.414 / 0.304 | 4.868 / 0.379 | 8.161 / 0.326 | 7.182 / 0.426 | 6.023 / 0.318 | 6.561 / 0.469 | 4.261 / 0.551 |
| FoldingNet (CVPR 2018) | 6.271 / 0.331 | 5.242 / 0.432 | 6.958 / 0.237 | 5.307 / 0.300 | 8.823 / 0.204 | 6.504 / 0.360 | 6.368 / 0.249 | 7.080 / 0.351 | 3.882 / 0.518 |
| PCN (3DV 2018) | 5.619 / 0.407 | 4.246 / 0.578 | 6.409 / 0.270 | 4.840 / 0.331 | 7.441 / 0.323 | 6.331 / 0.456 | 5.668 / 0.293 | 6.508 / 0.431 | 3.510 / 0.577 |
| TopNet (CVPR 2019) | 4.976 / 0.467 | 3.710 / 0.593 | 5.629 / 0.358 | 4.530 / 0.405 | 6.391 / 0.388 | 5.547 / 0.491 | 5.281 / 0.361 | 5.381 / 0.528 | 3.350 / 0.615 |
| PF-Net (CVPR 2020) | 3.873 / 0.551 | 2.515 / 0.551 | 4.453 / 0.399 | 3.602 / 0.453 | 4.478 / 0.489 | 5.185 / 0.559 | 4.113 / 0.409 | 3.838 / 0.614 | 2.871 / 0.656 |
| MSN (AAAI 2020) | 3.793 / 0.578 | 2.038 / 0.798 | 5.060 / 0.378 | 4.322 / 0.380 | 4.135 / 0.562 | 4.247 / 0.652 | 4.183 / 0.410 | 3.976 / 0.615 | 2.379 / 0.708 |
| GRNet (ECCV 2020) | 3.171 / 0.601 | 1.916 / 0.767 | 4.468 / 0.426 | 3.915 / 0.446 | 3.402 / 0.575 | 3.034 / 0.694 | 3.872 / 0.450 | 3.071 / 0.639 | 2.160 / 0.704 |
| PoinTr (ICCV 2021) | 2.851 / 0.683 | 1.686 / 0.842 | 4.001 / 0.516 | 3.203 / 0.545 | 3.111 / 0.662 | 2.928 / 0.742 | 3.507 / 0.547 | 2.845 / 0.723 | 1.737 / 0.780 |
| SeedFormer (ECCV 2022) | 2.902 / 0.688 | 1.716 / 0.835 | 4.049 / 0.551 | 3.392 / 0.544 | 3.151 / 0.668 | 3.226 / 0.777 | 3.603 / 0.555 | 2.803 / 0.716 | 1.679 / 0.786 |
| SDT (TVCG 2023) | 4.246 / 0.473 | 3.166 / 0.636 | 4.807 / 0.291 | 3.607 / 0.363 | 5.056 / 0.398 | 6.101 / 0.442 | 4.525 / 0.307 | 3.995 / 0.574 | 2.856 / 0.602 |
| SVDFormer (ICCV 2023) | 2.047 / 0.726 | 1.014 / 0.900 | 3.161 / 0.549 | 3.136 / 0.532 | 1.880 / 0.755 | 1.665 / 0.819 | 2.510 / 0.615 | 1.701 / 0.798 | 1.309 / 0.840 |
| ProxyFormer (CVPR 2023) | 2.734 / 0.628 | 1.086 / 0.890 | 3.033 / 0.540 | 3.151 / 0.490 | 4.001 / 0.506 | 2.256 / 0.701 | 3.493 / 0.474 | 3.263 / 0.631 | 1.592 / 0.795 |
| PointAttN (AAAI 2024) | 2.853 / 0.662 | 1.613 / 0.841 | 3.969 / 0.483 | 3.257 / 0.515 | 3.157 / 0.638 | 3.058 / 0.729 | 3.406 / 0.512 | 2.787 / 0.699 | 1.872 / 0.774 |
| Multi-modal Methods | | | | | | | | | |
| ViPC (CVPR 2021) | 3.308 / 0.591 | 1.760 / 0.803 | 4.558 / 0.451 | 3.183 / 0.512 | 2.476 / 0.529 | 2.867 / 0.706 | 4.481 / 0.434 | 4.990 / 0.594 | 2.197 / 0.730 |
| CSDN (TVCG 2024) | 2.570 / 0.695 | 1.251 / 0.862 | 3.670 / 0.548 | 2.977 / 0.560 | 2.835 / 0.669 | 2.554 / 0.761 | 3.240 / 0.557 | 2.575 / 0.729 | 1.742 / 0.782 |
| CDPNet (AAAI 2024) | 1.706 / 0.758 | 0.764 / 0.934 | 2.755 / 0.587 | 2.141 / 0.638 | 1.769 / 0.752 | 1.213 / 0.850 | 2.231 / 0.641 | 1.675 / 0.789 | 1.102 / 0.869 |
| XMFNet (NeurIPS 2022) | 1.443 / 0.796 | 0.572 / 0.961 | 1.980 / 0.662 | 1.754 / 0.691 | 1.403 / 0.809 | 1.810 / 0.792 | 1.702 / 0.723 | 1.386 / 0.830 | 0.945 / 0.901 |
| EGIINet (ECCV 2024) | 1.211 / 0.836 | 0.534 / 0.969 | 1.921 / 0.693 | 1.655 / 0.723 | **1.204 / 0.847** | 0.776 / 0.919 | 1.552 / 0.756 | **1.227 / 0.857** | 0.802 / 0.927 |
| I²GraphFormer (Ours) | **1.118 / 0.857** | **0.523 / 0.977** | **1.380 / 0.788** | **1.513 / 0.755** | 1.332 / 0.824 | **0.724 / 0.927** | **1.378 / 0.802** | 1.349 / 0.846 | **0.746 / 0.942** |

Table 1: Comparison of CD and F-Score under known categories on ShapeNet-ViPC.

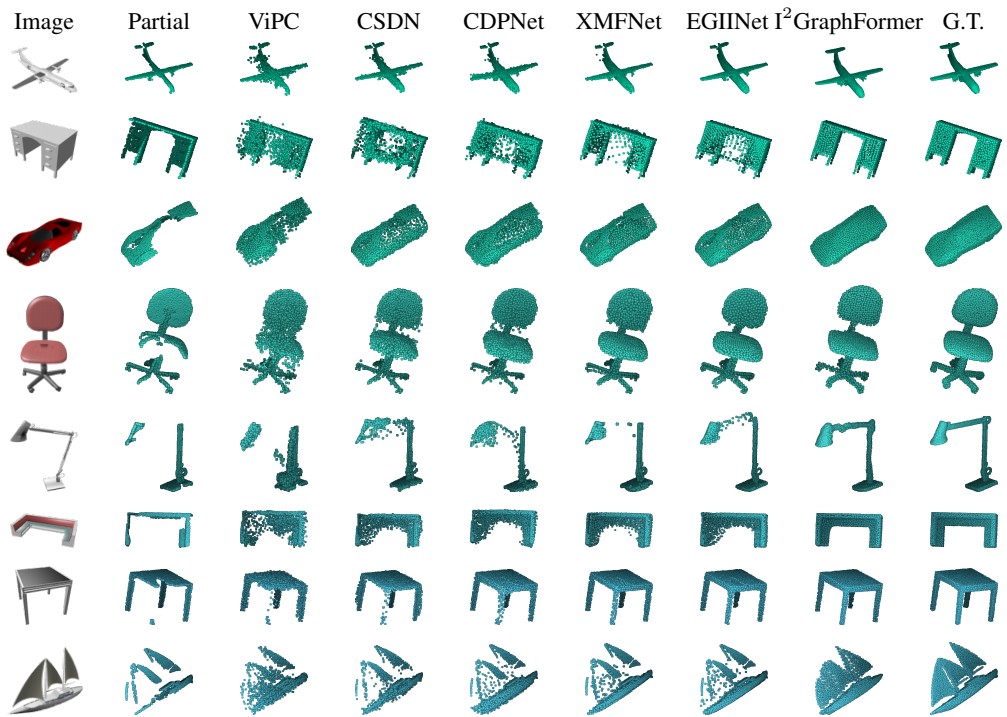

Figure 6: Comparison of visualization under known categories on ShapeNet-ViPC.

## 4.3 EVALUATION ON SHAPENET-VIPC

**Results on Known Categories.** In eight known categories, we compare our model with some representative multi-modal point cloud completion methods, such as ViPC Zhang et al. (2021), CSDN Zhu et al. (2024), CDPNet Du et al. (2024a), XMFNet Aiello et al. (2022), and EGIINet Xu et al. (2024), and some prominent single-modal point cloud completion methods, such as AtlasNet Groueix et al. (2018), FoldingNet Yang et al. (2018), PCN Yuan et al. (2018), TopNet Tchapmi et al. (2019), PF-Net Tchapmi et al. (2019), MSN Liu et al. (2020), GRNet Xie et al. (2020), PoinTr Yu et al. (2021), SeedFormer Zhou et al. (2022), SDT Zhang et al. (2023a), SVDFormer Zhu et al. (2023), ProxyFormer Li et al. (2023), and PointAttN Wang et al. (2024). Table 1 presents a comparison of quantitative experimental results between our method and other approaches. Across the eight known categories, our metrics are generally the best or second best. Moreover, the evaluation metrics show a significant improvement over those of the second-ranked method.

Figure 6 shows a qualitative comparison of our method with other multi-modal approaches. From the first row to the last, the categories represented are airplane, cabinet, car, chair, lamp, sofa, couch, and watercraft, respectively. Our method demonstrates superior performance in generating the global shape, particularly in the lamp category, where only our approach is able to produce a complete lamp arm. Furthermore, our method also excels in generating local details. Taking the couch and table categories as examples, our approach produces more compact surfaces with richer and more coherent details at corners and other intricate areas.

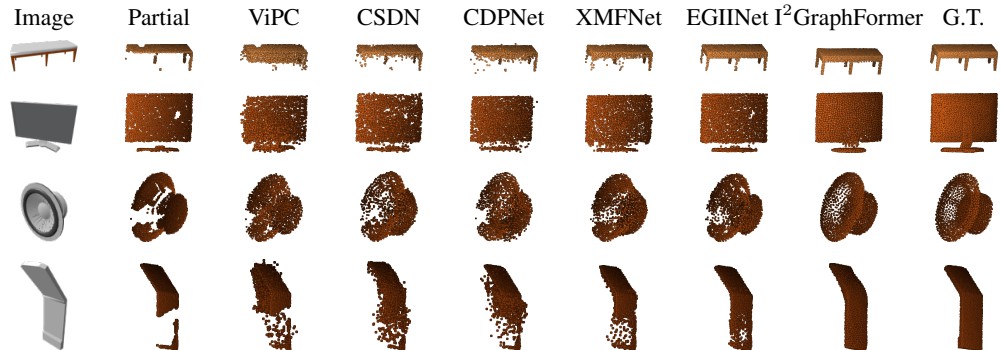

Figure 7: Comparison of visualization under unknown categories on ShapeNet-ViPC.

## 4.4 GENERALIZATION ABILITY EVALUATION

**Results on Unknown Categories of ShapeNet-ViPC.** In four known categories, we compare our model against some prominent multi-modal point cloud completion methods, such as ViPC Zhang et al. (2021), CDPNet Du et al. (2024a), CSDN Zhu et al. (2024), XMFNet Aiello et al. (2022), and EGIINet Xu et al. (2024), and some representative single-modal point cloud completion methods, such as PF-Net Huang et al. (2020), MSN Liu et al. (2020), GRNet Xie et al. (2020), PoinTr Yu et al. (2021), SeedFormer Zhou et al. (2022), SDT Zhang et al. (2023a), SVDFormer Zhu et al. (2023), ProxyFormer Li et al. (2023), and PointAttN Wang et al. (2024).

Table 2 presents a comparison of quantitative experimental results between our method and other approaches on four unknown categories. Our metrics are generally the best or second best. Moreover, the evaluation metrics show a significant improvement over those of the second-ranked method.

Figure 7 presents a qualitative comparison of our method with other multimodal approaches.

| Methods | CD ↓ / F-Score ↑ | | | | |
|---|---|---|---|---|---|
| | Avg | Bench | Monitor | Speaker | Cellphone |
| Single-modal Methods | | | | | |
| PF-Net (CVPR 2020) | 5.011 / 0.468 | 3.684 / 0.584 | 5.304 / 0.433 | 7.663 / 0.319 | 3.392 / 0.534 |
| MSN (AAAI 2020) | 4.684 / 0.533 | 2.613 / 0.706 | 4.818 / 0.527 | 8.259 / 0.291 | 3.047 / 0.607 |
| GRNet (ECCV 2020) | 4.096 / 0.548 | 2.367 / 0.711 | 4.102 / 0.537 | 6.493 / 0.376 | 3.422 / 0.569 |
| PoinTr (ICCV 2021) | 3.755 / 0.619 | 1.976 / 0.797 | 4.084 / 0.599 | 5.913 / 0.454 | 3.049 / 0.627 |
| SeedFormer (ECCV 2022) | 5.215 / 0.590 | 3.228 / 0.736 | 4.464 / 0.598 | 8.520 / 0.410 | 4.646 / 0.615 |
| SDT (TVCG 2023) | 6.001 / 0.327 | 4.096 / 0.479 | 6.222 / 0.268 | 9.499 / 0.197 | 4.189 / 0.362 |
| SVDFormer (ICCV 2023) | 4.414 / 0.541 | 2.785 / 0.703 | 4.451 / 0.524 | 7.196 / 0.367 | 3.225 / 0.571 |
| ProxyFormer (CVPR 2023) | 3.767 / 0.530 | 2.728 / 0.655 | 4.092 / 0.478 | 5.842 / 0.372 | 2.406 / 0.615 |
| PointAttN (AAAI 2024) | 3.674 / 0.605 | 2.135 / 0.764 | 3.741 / 0.591 | 5.973 / 0.428 | 2.848 / 0.637 |
| Multi-modal Methods | | | | | |
| ViPC (CVPR 2021) | 4.601 / 0.498 | 3.091 / 0.654 | 4.419 / 0.491 | 7.674 / 0.313 | 3.219 / 0.535 |
| CDPNet (AAAI 2024) | 4.462 / 0.589 | 3.122 / 0.714 | 4.100 / 0.593 | 7.611 / 0.418 | 3.013 / 0.629 |
| CSDN (TVCG 2024) | 3.656 / 0.631 | 1.834 / 0.798 | 4.115 / 0.598 | 5.690 / 0.485 | 2.985 / 0.644 |
| XMFNet (NeurIPS 2022) | 2.671 / 0.710 | 1.278 / 0.862 | 2.806 / 0.677 | 4.823 / 0.556 | 1.779 / 0.748 |
| EGIINet (ECCV 2024) | 2.354 / 0.750 | **1.047 / 0.902** | 2.513 / 0.716 | 4.282 / **0.591** | 1.575 / 0.792 |
| I²GraphFormer (Ours) | **1.735 / 0.781** | 1.232 / 0.869 | **1.679 / 0.777** | **2.879** / 0.590 | **1.150 / 0.887** |

Table 2: Comparison of CD and F-Score under unknown categories on ShapeNet-ViPC.

From the first row to the last, the categories represented are bench, monitor, speaker, and cellphone, respectively. The quantitative results indicate that our method achieves the best generalization performance on unseen categories, excelling in both global shape and local detail generation. For instance, in the bench, monitor, and cellphone categories, our method produces the smoothest and

Image   Partial   ViPC   CSDN   CDPNet   XMFNet   EGIINet I$^2$GraphFormer

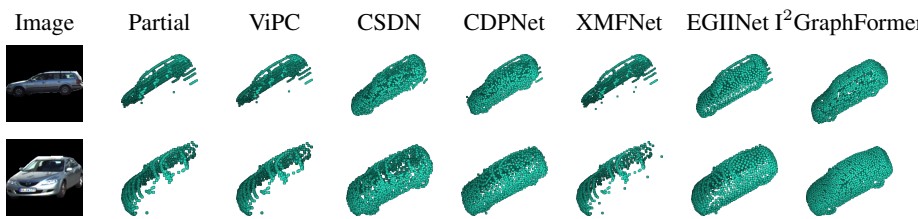

Figure 8: Comparison of visualization on KITTI.

most compact surfaces. In the speaker category, our approach is capable of generating circular shapes while capturing sufficient contour details.

**Results on Real-Scene KITTI.** We compared the qualitative generalization results of existing multi-modal methods, including ViPC Zhang et al. (2021), CDPNet Du et al. (2024a), CSDN Zhu et al. (2024), XMFNet Aiello et al. (2022), and EGIINet Xu et al. (2024), on the real-world KITTI dataset. We can only visually compare the completion results of different methods due to the absence of ground truth labels. The qualitative experimental results are shown in Figure 8, with each row representing a different car sample. These results indicate that, compared to other multi-modal methods, our approach performs better in recovering the global shape of unseen vehicles. Moreover, our method demonstrates superior ability in capturing fine details such as the car tires.

## 4.5 QUALITATIVE ANALYSIS OF INTER-GRAPH

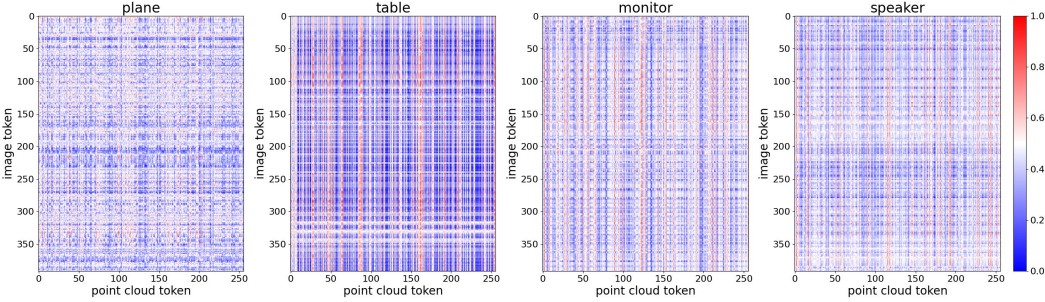

Figure 9: Visualization of the Inter-Graph Structure.

As shown in Figure 9, from left to right, the figures display the inter-graph weight maps for samples from the categories: plane, table, monitor, and speaker. In each visualization, the columns represent image tokens, while the rows correspond to point cloud tokens. The larger the value between each pair of point cloud token and image token, the more similar they are. Quantitative experimental results demonstrate that, when considering the relationship between the image modality and the point cloud modality, not all tokens should be interconnected. In other words, constructing a sparse graph structure more accurately reflects the complementary information inherent between the two modalities.

## 4.6 COMPLEXITY ANALYSES AND ABLATION STUDY

**Complexity Analyses.** In this part, we present a complexity analyses of I$^2$GraphFormer compared to existing multi-modal methods. Specifically, we evaluate the complexity of different approaches using two metrics, i.e., the number of parameters and model size. Table 3 summarizes the complexity analyses of various multi-modal methods. The results demonstrate that our method achieves the smallest parameter count and model size while delivering the best performance. Overall, our approach attains strong completion capabilities with low complexity.

| Methods | Params | FLOPs | Model Size | CD ↓ / F-Score ↑ | |
|---|---|---|---|---|---|
| | (M) ↓ | (G) ↓ | (MB) ↓ | Known | Unknown |
| ViPC (CVPR 2021) | 19.87 | 7.17 | 162.36 | 3.308 / 0.591 | 4.601 / 0.498 |
| CSDN (TVCG 2024) | 17.67 | 14.21 | 202.46 | 2.570 / 0.695 | 3.656 / 0.631 |
| CDPNet (AAAI 2024) | 13.77 | 6.04 | 143.63 | 1.706 / 0.758 | 4.462 / 0.589 |
| XMFNet (NeurIPS 2022) | 10.04 | 5.94 | 115.45 | 1.443 / 0.796 | 2.671 / 0.710 |
| EGIINet (ECCV 2024) | 9.47 | 5.96 | 108.94 | 1.211 / 0.836 | 2.354 / 0.750 |
| I²GraphFormer (Ours) | **8.63** | **5.53** | **98.97** | **1.118 / 0.857** | **1.735 / 0.781** |

Table 3: Multi-modal methods' complexity comparison.

**Ablation Study.** Table 4 presents the quantitative experimental results of the ablation study on the key components of I²GraphFormer. "w/o image modality" indicates that our model relies solely on point cloud input, excluding image data. "w/o intra-graph Transformer" means that our model removes the intra-graph Transformer component. "w/o inter-graph Transformer" indicates that our model removes the inter-graph Transformer component. "w/o DVGUM" indicates that our model replaces the DVGUM with the decoder of EGIINet

| Model | CD ↓ | F-Score ↑ |
|---|---|---|
| w/o image modality | 1.491 | 0.798 |
| w/o intra-graph Transformer | 1.503 | 0.803 |
| w/o inter-graph Transformer | 1.650 | 0.783 |
| w/o DVGUM | 2.124 | 0.720 |
| I²GraphFormer (Ours) | **1.118** | **0.857** |

Table 4: Ablation study on the components of I²GraphFormer.

Xu et al. (2024). The quantitative experimental results demonstrate that each component is crucial to our model. However, the decoder of EGIINet does not incorporate image information, which implies that the proposed Inter-Graph Transformer, intended to supplement point cloud information into image tokens, cannot be utilized by EGIINet's decoder. This limitation can be attributed to both the inherent design of EGIINet's decoder and the omission of image information during the decoding stage, leading to degraded performance.

## 5 LIMITATIONS

While our graph-based attention mechanisms are more efficient than fully-connected Transformers, they may still face scalability challenges when applied to extremely large scenes or high-resolution point clouds, limiting real-time applications. Our dual-view guided upsampling emphasizes geometric and visual cues but may still lack the ability for holistic scene-level semantic reasoning, which could be necessary for more comprehensive scene understanding tasks.

## 6 CONCLUSION AND DISCUSSIONS

We propose I²GraphFormer, a novel approach for multi-modal point cloud completion that demonstrates high-efficiency performance in both synthetic and real-world scenarios. By deeply capturing feature relations within point cloud and image modalities from an intra- and inter-graph perspective, our method effectively models complex geometric and semantic correlations to facilitate complementary information exchange and cross-modal fusion. Additionally, we introduce a dual-view guided upsampling module that directs the reconstruction process using both geometric and image cues, enabling the generation of finer-grained and more accurate point clouds.

Future work will extend image integration to more complex semantic scene completion tasks, aiming for comprehensive reconstruction and semantic understanding in multi-object environments. By exploiting complementary multi-modal data, the model's multi-object recognition and segmentation capabilities can be improved, leading to more precise and semantically rich scene reconstructions.

### REPRODUCIBILITY STATEMENT

The dataset utilized in this study is consistent with that used in previously proposed methods, and no additional special preprocessing steps have been applied.

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

# A APPENDIX

## A.1 THE USE OF LARGE LANGUAGE MODELS (LLMS)

We used a large language model (LLM) to help polish the submitted manuscript.

