# OpenReview forum: "Multi-Modal Point Cloud Completion with Intra- and Inter-Graph Transformer"
_ICLR.cc/2026/Conference — Submitted to ICLR 2026_

### Official Review · Reviewer_hdKR · 2025-10-18

**Soundness:** 3
**Presentation:** 3
**Contribution:** 2
**Rating:** 4
**Confidence:** 5

**Summary:**

This paper proposes I² GraphFormer, a multi-modal point cloud completion framework leveraging sparse intra- and inter-graph attention in Transformer-based architectures. The approach encodes both images and point clouds and constrains attention to local neighborhoods within and across modalities, aiming for improved efficiency and interaction expressiveness. Experiments on the ShapeNet-ViPC benchmark and KITTI dataset demonstrate competitive quantitative and qualitative results, with additional ablation studies highlighting the roles of each module in the architecture.

**Strengths:**

- The proposed method significantly outperforms existing methods on widely-used datasets.
- Overall, the manuscript is well-written.

**Weaknesses:**

- Using cross attention for modality fusion has been widely explored and proved effective. The motivation for using graphs in I2GraphFormer, as opposed to using traditional Transformers, is not fully clear in the paper.
- Using feature distance to build graphs can be computationally expensive.
- There is no fundamental novelty regarding the cross-modal graph fusion.
- Recently, a trend of zero-shot point completion using multi-modal generative models has emerged (e.g., SDSComplete—NeurIPS 2024, ComPC—ICLR 2025, GenPC—CVPR 2025). These methods should at least be discussed in the related work section.
- The manuscript lacks an analysis of failure cases and limitations.

**Questions:**

Please see the weaknesses.

---

> ### Author Response · Authors · 2025-11-20
> **Thank you for your valuable suggestions and comments.**
>
> Thank you for your valuable suggestions and comments.
> >Q1: Our graph-based design offers advantages in modeling selective local relationships and sparse connections. By introducing neighborhood constraints, we reduce redundant computations and improve efficiency. Additionally, the graph structure enables more intuitive modeling of local geometric and semantic relationships, enhancing representation. Our sparse connectivity ensures the model focuses on relevant neighbors, balancing performance and computational cost. **Strength 2 of Reviewer yMF8 also conveyed a similar perspective.**
>
> >Q2: We acknowledge that building graphs based on feature distances can be expensive. To address this, we implement a top-K local neighbor sampling strategy, limiting calculations to the most relevant neighbors for each node. This sparsification significantly reduces computational load while maintaining effectiveness. To address your concerns, we investigated the impact of different K values on the model's performance. The quantitative results are presented in the table below. The experimental results demonstrate that **a higher number of edges corresponds to increased model complexity. However, this does not necessarily lead to optimal performance**. This further corroborates our main idea that constructing sparse graph structure within the Transformer framework enables more efficient intra-modal feature mining and inter-modal interactions.
> >
> >|Model|Parameters (M)↓|FLOPs (G)↓|Model Size (MB)↓|CD ↓ / F-Score ↓|
> >|-|-|-|-|-|
> >|K=12|8.61|4.93|98.78|1.401 / 0.781|
> >|K=16|8.63|5.53|98.97|**1.380** / **0.788**|
> >|K=20|8.64|5.86|99.03|1.391 / 0.784|
> >|fully-connect|16.56|12.03|189.86|1.397 / 0.782|
>
> >Q3: Regarding the cross-modal graph fusion, our main contribution is the first to introduce intra- and inter-graph Transformer within multi-modal point cloud completion. This dual-perspective graph modeling captures both modality-specific and cross-modal relationships explicitly, offering a novel and effective framework. Unlike existing methods, our approach leverages intra-graph and inter-graph fusion to better exploit modality interactions, representing a significant novelty in this field.
>
> >Q4: We have added a discussion on these zero-shot point cloud completion methods in the revised version, see Line 106 on Page 2 of the revised manuscript.
>
> >Q5: We have added a discussion on Limitations in the revised version, see Line 513 on Page 10 of the revised manuscript.

---

### Official Review · Reviewer_D8jU · 2025-10-27

**Soundness:** 2
**Presentation:** 3
**Contribution:** 3
**Rating:** 4
**Confidence:** 3

**Summary:**

This paper introduces an efficient multi-modal point cloud completion framework. By incorporating an Intra- and Inter-Graph Transformer structure with sparse graph connections, the method enables effective information interaction between point cloud and image modalities, capturing complex geometric and semantic relationships. Furthermore, a dual-view guided upsampling module is proposed to refine the reconstruction process using both geometric and image cues, resulting in finer and more accurate point clouds. Extensive experiments on synthetic and real-world datasets demonstrate that the proposed model outperforms existing multi-modal approaches across various evaluation scenarios.

**Strengths:**

- The paper is easy to follow.
- The proposed method demonstrates excellent performance on the cross-modal point cloud completion task.
- The authors identify the efficiency limitations of existing Transformer-based point cloud completion methods and explore an effective and efficient solution to address this issue.

**Weaknesses:**

**Major Concerns**
- The paper briefly mentions in line 63 that Transformers incur high computational cost, which is understandable. However, it is unclear why the Transformer "fails to adequately capture the complex relations between modalities." This claim lacks sufficient justification and should be further elaborated.
- The proposed intra-graph and inter-graph structures do not seem to follow conventional graph formulations. Each node is encoded by three distinct MLP layers, resulting in three feature representations. The method computes similarities between nodes from Q and K, selects the top-k similar nodes to construct adjacency relationships between Q and K, and then aggregates corresponding V nodes based on this relation. However, this adjacency only reflects the relationship between Q and K, seemingly unrelated to V.
From a graph-based perspective, this makes the structure somewhat unconventional. While QKV MLPs remain essential to Transformer architectures, the proposed design appears to be a more efficient variant of a Transformer block rather than a novel graph model. Despite its strong empirical performance, the paper lacks a clear explanation of why this structure leads to such improvements. The authors are encouraged to provide deeper analysis or intuition behind the observed performance gains.
- Several critical settings are missing. For instance:
  - What is the value of k in the top-k selection?
  - How many layers are used for the intra-graph Transformer?
  - Is the inter-graph Transformer a single layer?
  - What are the specific details of the geometry-guided part in the Dual-View Guided Upsampling Module? According to the ablation study, removing this module causes a significant performance drop, suggesting it plays a more important role than the intra- and inter-graph components highlighted as the paper's main contribution. However, this module is insufficiently described and not emphasized in the introduction. The authors should clarify its design and significance.
- It would be valuable to analyze the effect of different k values on performance and to include a comparison without top-k selection (i.e., using all tokens) to better understand its impact.

**Minor Concerns**
- For computational efficiency analysis, it is recommended to include MACs and FLOPs metrics.
- Figure 1 is not referenced in the main text and should be properly cited.

**Questions:**

Please refer to the Weaknesses part.

---

> ### Author Response · Authors · 2025-11-20
> **We sincerely appreciate your careful reading and valuable suggestions, and we hope to address your concerns.**
>
> We sincerely appreciate your careful reading and valuable suggestions, and we hope to address your concerns.
> >Q1: As noted by Reviewer yMF8: "**The core idea of replacing fully-connected cross-modal attention with a sparse graph-based paradigm is well-motivated. The explicit separation into intra-graph and inter-graph learning provides a structured and interpretable framework for modality interaction that differs significantly from prior work (Strength 2).**" Existing fully-connected attention-based Transformer methods, which consider intra-modal and inter-modal interactions, entail redundancy. As stated in our paper: "not all local tokens are similar and therefore do not all have edges connecting them." To alleviate your concerns, we have conducted a qualitative analysis of the inter-graph structure. The detailed results can be found on Line 455 of Page 9 in the revised version. Our final conclusion is that constructing a sparse graph structure more accurately reflects the complementary information inherent between the two modalities.
>
> >Q2: Indeed, I$^{2}$GraphFormer is not a traditional graph-based model; rather, it is a more efficient Transformer model that incorporates the sparse connectivity principles derived from graph models. In the revised version (Section 4.5 in Line 455 on Page 9 of the revised version), we performed a qualitative examination of the inter-graph structure, ultimately concluding that a sparse graph structure better captures the complementary information inherent between the two modalities.
>
> >Q3:
> >* In the paper, the value of k is set to 16.
> >* Two layers are used for the intra-graph Transformer, each dedicated to internal information aggregation within the image and point cloud modalities, respectively.
> >* Yes, the paper employs two inter-graph Transformers, each serving to supplement image tokens with point cloud information and to incorporate image information into point cloud tokens, respectively.
> >* (1) The geometry guided branch constructs a top-K graph structure based on the point cloud coordinates, and subsequently aggregates the point cloud tokens using this graph structure to obtain refined point cloud tokens. This description has been added to Line 256 on Page 5 of the revised version. (2) In the ablation study, we stated: ``w/o DVGUM indicates that our model replaces the DVGUM with the decoder of EGIINet [1].'' However, the decoder of EGIINet does not incorporate image information, which implies that the proposed Inter-Graph Transformer, intended to supplement point cloud information into image tokens, cannot be utilized by EGIINet's decoder. This limitation can be attributed to both **the inherent design of EGIINet's decoder** and **the omission of image information during the decoding stage**, leading to degraded performance. The relevant description has also been added to Line 507 on Page 10 of the revised paper. (3) We also emphasized the role of DVGUM in the Introduction, as seen in Line 74 on Page 2 of the revised version.
>
> > [1] Hang Xu, et. al. Explicitly guided information interaction network for cross-modal point cloud completion. ECCV, 2024.
>
> >Q4: Due to time constraints, we only discussed the impact of varying K values on model performance within the cabinet category, including the scenario where all tokens are used without applying Top-k selection (i.e., fully-connect). The quantitative experimental results are summarized in the table below. The experimental results demonstrate that **a higher number of edges corresponds to increased model complexity. However, this does not necessarily lead to optimal performance**. This further corroborates our main idea that constructing sparse graph structure within the Transformer framework enables more efficient intra-modal feature mining and inter-modal interactions.
> >
> >|Model|Parameters (M)↓|FLOPs (G)↓|Model Size (MB)↓|CD ↓ / F-Score ↓|
> >|-|-|-|-|-|
> >|K=12|8.61|4.93|98.78|1.401 / 0.781|
> >|K=16|8.63|5.53|98.97|**1.380** / **0.788**|
> >|K=20|8.64|5.86|99.03|1.391 / 0.784|
> >|fully-connect|16.56|12.03|189.86|1.397 / 0.782|
>
> >Q5: We have introduced the FLOPs metric in addition to the previously presented complexity comparisons. The quantitative results are presented in the table (Table 3 on Page 10 of the revised version) below.
> >
> >|Methods|Parameters (M)↓|FLOPs (G)↓|Model Size (MB)↓|Known (CD ↓ / F-Score ↓)|Novel (CD ↓ / F-Score ↓)|
> >|-|-|-|-|-|-|
> >|ViPC (CVPR 2021)|19.87|7.17|162.36|3.308 / 0.591|4.601 / 0.498|
> >|CSDN (TVCG 2024)|17.67|14.21|202.46|2.570 / 0.695|3.656 / 0.631|
> >|CDPNet (AAAI 2024)|13.77|6.04|143.63|1.706 / 0.758|4.462 / 0.589|
> >|XMFNet (NeurIPS 2022)|10.04|5.94|115.45|1.443 / 0.796|2.671 / 0.710|
> >|EGIINet (ECCV 2024)|9.47|5.96|108.94|1.211 / 0.836|2.354 / 0.750|
> >|I$^{2}$GraphFormer (Ours)|**8.63**|**5.53**|**98.97**|**1.118** / **0.857**|**1.735** / **0.781**|
>
> >Q6: Thank you for noting that Figure 1 was not referenced. We have now cited it appropriately in the revised manuscript to enhance clarity.

---

### Official Review · Reviewer_yMF8 · 2025-10-31

**Soundness:** 3
**Presentation:** 3
**Contribution:** 3
**Rating:** 6
**Confidence:** 5

**Summary:**

This paper introduces I2GraphFormer, a novel method for multi-modal point cloud completion that leverages complementary image information. The key contribution is a graph-based Transformer architecture designed to address the high computational cost and limited expressiveness of fully-connected attention in existing methods. The model operates by first encoding point clouds and images into tokens, then processing them through two main stages: (1) Intra-Graph Transformers that capture internal structures within each modality using sparse, top-K attention, and (2) Inter-Graph Transformers that facilitate cross-modal information exchange via a bipartite graph structure. Finally, a Dual-View Guided Upsampling Module reconstructs the complete point cloud using guidance from both geometric and semantic perspectives. The authors demonstrate state-of-the-art performance on the ShapeNet-ViPC dataset and the real-world KITTI benchmark, while also achieving lower model complexity than competitors.

**Strengths:**

1. Multi-modal completion is a task of considerable practical importance autonomous driving. By offering a more efficient and expressive model for fusing point cloud and image data, this work represents a meaningful advancement. The demonstrated performance on real-world KITTI data underscores its potential for practical application.
2. The core idea of replacing fully-connected cross-modal attention with a sparse graph-based paradigm is well-motivated. The explicit separation into intra-graph and inter-graph learning provides a structured and interpretable framework for modality interaction that differs significantly from prior work.
3. The paper is technically sound. The experimental evaluation is thorough, encompassing standard benchmarks (ShapeNet-ViPC), generalization to unseen categories, and real-world data (KITTI). The results show good performance in both completion quality and model efficiency.

**Weaknesses:**

1. Scalability.  While the top-K graph attention reduces complexity, its scalability to very large token sets or higher-resolution inputs is not thoroughly analyzed. The computational cost of the pairwise cosine similarity calculation and top-K selection for all queries, though less than full attention, could still become a bottleneck. An analysis of how the method's time/memory consumption scales with the number of tokens (N, M) and the choice of K would strengthen the paper.

2. Insight of token fusion. The paper shows that the model works well, but could provide more insight into what is being transferred between modalities. A qualitative analysis or visualization of which image tokens attend to which point cloud tokens (and vice-versa) in the inter-graph transformer would be very valuable. For instance, does the image primarily provide high-level semantic context, or does it also help infer fine-grained geometric details? This would deepen the understanding of the cross-modal fusion process.
3. The contribution of the image. The paper clearly demonstrates that the full multi-modal model performs well. However, it lacks an ablation study that quantifies the specific contribution of the single-view image to the final completion result. This would clearly isolate the performance gain attributable to the image information, both in known and unknown categories, and strengthen the claim that the image provides essential complementary guidance.
4. Robustness. The method assumes well-aligned image and point cloud pairs. A key limitation in real-world systems is imperfect calibration or temporal misalignment between sensors. The paper should evaluate the robustness of I2GraphFormer to such misalignments.

**Questions:**

Scalability: Could the authors provide a more detailed complexity analysis (e.g., FLOPs or memory usage vs. token count) for the graph construction and attention steps compared to a standard transformer? Have you experimented with larger token sizes or values of K, and what were the observed trade-offs?

Interpretation: Can you provide a qualitative visualization or case study showing examples of the top-K connections formed in the inter-graph transformer? For example, when a point cloud token from a missing car part queries the image, which image regions are most frequently attended to? This would help clarify the nature of the cross-modal complementarity.

---

> ### Author Response · Authors · 2025-11-20
> **We sincerely appreciate your positive feedback and valuable suggestions.**
>
> We sincerely appreciate your positive feedback and valuable suggestions.
> >Q1: Indeed, our method's scalability concerning extremely large token sizes has not been thoroughly analyzed. This is primarily because the point cloud resolution in current multi-modal point cloud completion benchmarks is typically around 2048, which is not particularly high. To address your concern to some extent, we have examined the impact of different K values on the completion performance. The quantitative results are presented in the table below. The experimental results demonstrate that **a higher number of edges corresponds to increased model complexity. However, this does not necessarily lead to optimal performance**. This further corroborates our main idea that constructing sparse graph structure within the Transformer framework enables more efficient intra-modal feature mining and inter-modal interactions.
> >
> >|Model|Parameters (M)↓|FLOPs (G)↓|Model Size (MB)↓|CD ↓ / F-Score ↓|
> >|-|-|-|-|-|
> >|K=12|8.61|4.93|98.78|1.401 / 0.781|
> >|K=16|8.63|5.53|98.97|**1.380** / **0.788**|
> >|K=20|8.64|5.86|99.03|1.391 / 0.784|
> >|fully-connect|16.56|12.03|189.86|1.397 / 0.782|
>
> >Q2: In the revised version, we discussed the qualitative analysis of the inter-graph structure (Line 455 on Page 9). In each visualization, the columns represent image tokens, while the rows correspond to point cloud tokens. The larger the value between each pair of point cloud token and image token, the more similar they are. Quantitative experimental results demonstrate that, when considering the relationship between the image modality and the point cloud modality, not all tokens should be interconnected. In other words, constructing a sparse graph structure more accurately reflects the complementary information inherent between the two modalities.
>
> >Q3: We have included this ablation study in the revised version, which can be found in Line 499 on Page 10 of the updated manuscript. To facilitate, we will include the results below for your reference.
> >
> >|Model|CD ↓ | F-Score ↓|
> >|-|-|-|
> >|w/o image modality|1.491|0.798|
> >|I$^{2}$GraphFormer (Ours)|**1.118**|**0.857**|
>
>
> >Q4: It should be noted that in multi-modal point cloud completion, the inputs consist of a complete image and a partially occluded point cloud. The designed model is expected to generate a complete point cloud. Therefore, the information completeness of the two modalities is inconsistent, and they cannot be considered as aligned or paired in this context. The issue you raised is indeed of significant practical importance, and we intend to focus on addressing it in our future research.
>
> >Q5: To address your concerns, we investigated the impact of different K values on the model's performance. The quantitative results are presented in the table below. The experimental results demonstrate that **a higher number of edges corresponds to increased model complexity. However, this does not necessarily lead to optimal performance**. This further corroborates our main idea that constructing sparse graph structure within the Transformer framework enables more efficient intra-modal feature mining and inter-modal interactions.
> >
> >|Model|Parameters (M)↓|FLOPs (G)↓|Model Size (MB)↓|CD ↓ / F-Score ↓|
> >|-|-|-|-|-|
> >|K=12|8.61|4.93|98.78|1.401 / 0.781|
> >|K=16|8.63|5.53|98.97|**1.380** / **0.788**|
> >|K=20|8.64|5.86|99.03|1.391 / 0.784|
> >|fully-connect|16.56|12.03|189.86|1.397 / 0.782|
>
> >Q6: What we need to clarify is that, within the inter-graph structure, there are typically a large number of point cloud tokens and image tokens. For example, in Section 4.5 of the revised version (Line 455 on Page 9), it can be intuitively observed that the majority of similarities between the point cloud and image tokens are relatively small. This observation, to some extent, supports the rationality of the top-K graph construction.

---

### Official Review · Reviewer_fngW · 2025-10-31

**Soundness:** 2
**Presentation:** 1
**Contribution:** 2
**Rating:** 2
**Confidence:** 3

**Summary:**

This paper proposes I2GraphFormer, an Intra- and Inter-Graph Transformer for multi-modal point cloud completion, which aims to leverage complementary image information to enhance 3D completion quality.

**Strengths:**

+ The proposed method has a smaller model size compared to existing models.
+ The claim of outperforming SOTA methods with lower complexity suggests empirical validation.

**Weaknesses:**

- Unclear novelty. The paper does not clearly differentiate I2GraphFormer from prior graph-based or cross-modal attention methods (e.g., Graphormer, CrossGraphFusion). The novelty appears incremental unless stronger distinctions or innovations are articulated.

- Insufficient motivation and justification. The rationale for adopting the intra- and inter-graph design is not well explained. It remains unclear why this design is superior to existing approaches. Moreover, while the sparse graph mechanism is intuitive, the paper lacks theoretical analysis or ablation studies demonstrating why inter-graph connections contribute to better completion quality.

- Weak support for the “low complexity” claim. The paper repeatedly emphasizes its low complexity, but the evaluation only compares model size without providing concrete analyses such as computational complexity (e.g., FLOPs) or actual runtime measurements.

- Unclear motivation for the Dual-View Guided Upsampling Module. The purpose and necessity of this module are not well justified. The paper does not explain why this specific design is required。

**Questions:**

See the Weaknesses.

---

> ### Author Response · Authors · 2025-11-20
> **We sincerely appreciate your valuable feedback and hope to address your concerns.**
>
> We sincerely appreciate your valuable feedback and hope to address your concerns.
> >Q1: We would like to clarify that previous graph Transformer methods, such as Graphormer, **primarily target graph-structured data objects**, whereas our I$^{2}$GraphFormer is specifically **designed for 3D point cloud data**. Our main contribution is the first to explicitly introduce intra- and inter-graph transformers architectures tailored for multi-modal point cloud completion, facilitating more efficient mutual information enrichment across modalities.
>
> >Q2: **Strength 2 of Reviewer yMF8 states: "The core idea of replacing fully-connected cross-modal attention with a sparse graph-based paradigm is well-motivated. The explicit separation into intra-graph and inter-graph learning provides a structured and interpretable framework for modality interaction that differs significantly from prior work."**
> >* Our approach is motivated by the need to better capture both local structural information within each modality (via intra-graph connections) and cross-modal relationships (via inter-graph connections). This design allows for more targeted and efficient information exchange compared to traditional methods that often rely on dense attention mechanisms or simple feature concatenation, which can be less effective and computationally more expensive.
> >* Regarding the sparse graph mechanism, we agree that its advantages are intuitive; however, as our ablation studies (Page 10) demonstrate, the inter-graph connections significantly improve completion quality by facilitating meaningful cross-modal interactions while maintaining computational efficiency. These findings are supported by experiments showing that our graph-based design better enhances the mutual compensation between modalities, leading to superior performance.
>
> >Q3: To fully address your concerns, we have introduced the FLOPs metric in addition to the previously presented complexity comparisons. The quantitative results are presented in the table (Table 3 on Page 10 of the revised version) below.
> >
> >|Methods|Parameters (M)↓|FLOPs (G)↓|Model Size (MB)↓|Known (CD ↓ / F-Score ↓)|Novel (CD ↓ / F-Score ↓)|
> >|-|-|-|-|-|-|
> >|ViPC (CVPR 2021)|19.87|7.17|162.36|3.308 / 0.591|4.601 / 0.498|
> >|CSDN (TVCG 2024)|17.67|14.21|202.46|2.570 / 0.695|3.656 / 0.631|
> >|CDPNet (AAAI 2024)|13.77|6.04|143.63|1.706 / 0.758|4.462 / 0.589|
> >|XMFNet (NeurIPS 2022)|10.04|5.94|115.45|1.443 / 0.796|2.671 / 0.710|
> >|EGIINet (ECCV 2024)|9.47|5.96|108.94|1.211 / 0.836|2.354 / 0.750|
> >|I$^{2}$GraphFormer (Ours)|**8.63**|**5.53**|**98.97**|**1.118** / **0.857**|**1.735** / **0.781**|
>
> >Q4: The Dual-View Guided Upsampling Module aims to decode the final completed results by utilizing mutually complementary point cloud and image tokens. A straightforward explanation is that **we need to design a module that uses point cloud and image tokens to generate the final completion result**. Specifically, in the geometry-guided branch, low-resolution point cloud coordinates and point cloud tokens are employed to decode refined point cloud features. In the image-guided branch, image tokens are integrated into the point cloud tokens. Subsequently, these enriched point cloud features are processed through an MLP to generate the final high-resolution point cloud coordinates.

---

### Meta-Review · Area_Chair_kPjR · 2025-12-24

**Summary:**

This paper studies multi-modal point cloud completion and proposes I²GraphFormer, which replaces fully-connected cross-modal attention with sparse intra-graph and inter-graph Transformer blocks, plus a dual-view guided upsampling module (DVGUM). Reviewers agree the paper targets an important problem and reports strong results on ShapeNet-ViPC and KITTI, with an efficiency-oriented design. However, after considering the rebuttal and post-discussion content, substantial concerns remain about (i) the clarity and strength of the methodological novelty over prior cross-modal attention / graph-transformer fusion, (ii) the sufficiency of motivation/analysis explaining why the proposed graph construction and intra/inter decomposition are necessary, and (iii) incomplete evidence for the “low complexity / scalability / robustness” claims beyond added FLOPs and limited ablations. In particular, while the rebuttal adds FLOPs comparisons and component ablations, key doubts about whether the approach is fundamentally beyond a more efficient Transformer variant, and whether it is robust/scalable in practical multimodal settings (e.g., misalignment), are not fully resolved.

**Reviewer Concerns:**

Reviewer fngW: Despite added clarifications, the paper still does not convincingly distinguish I²GraphFormer from prior graph/cross-modal attention methods, and the justification for the intra/inter-graph design and DVGUM remains weak.

Reviewer yMF8: The approach is promising, but concerns about scalability/complexity analysis, interpretability of token fusion, and robustness to real-world misalignment are only partially addressed (with some deferred to future work).

Reviewer D8jU: The method’s “graph” formulation and why it yields gains remain insufficiently explained, and key implementation details/analyses were initially missing; rebuttal fills some settings but deeper justification is still lacking.

Reviewer hdKR: The work is well written with strong results, but the motivation/novelty of graph-based fusion over standard cross-attention is unclear, graph construction can be expensive, and broader related-work/limitations discussion was initially missing (partially patched in rebuttal).

**Reviewer Scores:**

Reviewer fngW: Likely no change (rated 2: reject).

Reviewer yMF8: Likely no change or a slight increase (rated 6: marginally above acceptance threshold).

Reviewer D8jU: Likely no change (rated 4: marginally below acceptance threshold).

Reviewer hdKR: Likely no change (rated 4: marginally below acceptance threshold).

---

### Decision · Program_Chairs · 2026-01-26

Reject